# Chemokines and Cytokines Profiles in Patients with Antineutrophil Cytoplasmic Antibodies-Associated Vasculitis: A Preliminary Study

**DOI:** 10.3390/ijms242015319

**Published:** 2023-10-18

**Authors:** Agnieszka Daca, Hanna Storoniak, Alicja Dębska-Ślizień, Mariusz Andrzej Kusztal, Magdalena Krajewska, Katarzyna Aleksandra Lisowska

**Affiliations:** 1Department of Pathophysiology, Medical University of Gdańsk, 80-211 Gdansk, Poland; katarzyna.lisowska@gumed.edu.pl; 2Department of Nephrology, Transplantology, and Internal Diseases, Medical University of Gdańsk, 80-211 Gdansk, Poland; hstoroniak@gumed.edu.pl (H.S.); alicja.debska-slizien@gumed.edu.pl (A.D.-Ś.); 3Department of Nephrology and Translational Medicine, Medical University of Wrocław, 50-137 Wroclaw, Poland; mariusz.kusztal@umw.edu.pl (M.A.K.); magdalena.krajewska@umw.edu.pl (M.K.)

**Keywords:** ANCA-associated vasculitis, vessels, inflammation, cytokines, chemokines

## Abstract

The damage to small vessels in AAV and inflammatory reactions are accompanied by the release of various chemokines and cytokines. Using a flow cytometry technique, we assessed the levels of specific cytokines, namely IL-1β IL-6, IL-8, IL-10, IL12p70, and TNF, and chemokines, IFN-α, IP-10, and MIG in the serum from 9 healthy volunteers and 20 AAV patients, where 11 of the patients were not treated and evaluated at the time of diagnosis and 9 were already diagnosed and taking CY + GCS. The obtained results were then compared considering the activity of the disease, the type and titre of the ANCA antibodies, the inflammatory status, and the kidneys’ condition. Amongst others, the IL-6, IL-8, IL-10, TNF, and MIG levels were much higher in the serum of AAV patients than in healthy controls, whereas the level of IL-1β was higher in healthy volunteers. Additionally, the levels of IL-6, IL-10, IP-10, and MIG negatively correlated with the eGFR level, while the level of IFN-α positively correlated with the titre of PR3-ANCA. As most of the molecules are implicated in trafficking primed neutrophils towards small vessels, looking for links between the levels of these cytokines/chemokines and the clinical symptoms of AAV may facilitate the diagnosis and predict the progression of the disease.

## 1. Introduction

Cytokines and chemokines play an important role in the human body, regulating such fundamental processes as, for e.g., embryonic development, angiogenesis, metabolism, immunity, and ageing. Their involvement in various pathologies also cannot be ignored. They mediate acute and chronic inflammatory responses, autoimmune and autoinflammatory reactions, and many others [1]. It would be difficult to pinpoint any activity within the living body independent from the complicated net of cytokines and chemokines released by immune and non-immune cells. Antineutrophil cytoplasmic antibodies (ANCA)-associated vasculitis (AAV) is a group of relatively rare diseases involving the presence of a wide variety of symptoms, all linked with inflammatory cells infiltration of small vessels leading to necrotic changes [2]. The primary division of AAV includes three forms, namely GPA (granulomatosis with polyangiitis), MPA (microscopic polyangiitis), and EGPA (eosinophilic granulomatosis with polyangiitis), where GPA is the most common, and EGPA is the least common [2,3]. Even though there are suggestions that the division based on ANCA specificity (PR3—proteinase 3-ANCA vs. MPO—myeloperoxidase-ANCA) would be better, as the presence of specific antibodies indicates certain features of the disease itself, for e.g., the relapse rate is higher in PR3-ANCA, and the mortality rate is higher in MPO-ANCA, the current classification proposed by the EULAR/ACR (European Alliance of Associations for Rheumatology/American College of Rheumatology) is still based on the clinical phenotype [4,5,6,7].

As the affected elements in AAV are mainly small vessels, the heterogeneity of possible symptoms is relatively high, starting from a localised skin rash and finishing with fulminant multisystem disease, where the symptoms will be partially linked with a specific form of vasculitis [2,6].

ANCA-associated antibodies recognise their targets on the surface of patients’ monocytes and neutrophils. The binding causes the release of PR3 and MPO, and various chemokines and cytokines [8]. Based on immunofluorescence patterns, PR3-ANCA shows cytoplasmic staining, which is why they are called c-ANCA, while MPO-ANCA shows perinuclear staining (p-ANCA) [9]. Apart from the pro-inflammatory action of cytokines, the released PR3 and MPO are internalised by the endothelial cells causing the activation of other specific processes. PR3 activates the apoptosis of endothelial cells, and MPO leads to the production of oxidants [10,11]. All of these phenomena lead to the damage of small vessels. Additionally, primed neutrophils bind to the vessel wall’s surface, leading to further aggravation of the inflammatory reaction by again releasing chemokines and cytokines [10,12]. The released chemokines and cytokines, e.g., TNF-α (tumour necrosis factor α), IL-1 (interleukin 1), IL-8, and MCP (monocyte chemoattractant protein), attract other immune cells to the site of the reaction, which in turn start to release their own set of chemokines and cytokines, accelerating in that way the deterioration of the inflammatory processes in the small vessels [13].

Due to its rarity, AAV is frequently misdiagnosed as infections, malignancies, depression, or osteoarthritis, especially in older patients. Due to the variety of affected organs and heterogeneous symptoms, overlapping other diseases, AAV can remain undiagnosed for months or years until ANCA testing is performed. Given the rarity of AAV and the existence of diseases that mimic vasculitis, diagnosis should be reviewed periodically, especially in cases of inadequate response to treatment or inconsistency of some of the disease manifestations with AAV. Currently, the primary diagnostic scheme relies on clinical features in combination with positive ANCA serology. Importantly, positive ANCA serology can also be found in other conditions with systemic symptoms or asymptomatic patients. ANCA testing is also utilised for patient management monitoring and relapse prevention. However, the role of serial ANCA measurement in determining treatment during remission remains controversial, thus urging the need for new clinically valuable biomarkers. An example of such biomarkers that can be easily measured is cytokines and chemokines, which play an important role in inflammation in AAV.

There are different articles describing changes in cytokines and/or chemokines in AAV patients [14]. The problem is that it is difficult to compare the results due to the use of different scientific methods to establish cytokine levels, different materials used in the studies (serum vs. plasma), different study models, and different study groups (studies showing results only for treated patients or only patients without treatment etc.). Therefore, we aimed to assess the levels of specific cytokines, namely IL-1β IL-6, IL-8, IL-10, IL12p70, and TNF, and chemokines, IFN-α (interferon α), IP-10 (interferon γ-induced protein 10 or CXCL10), and MIG (monokine induced by γ or CXCL9) in the serum of AAV patients, taking into account various factors that could influence the levels of the cytokines tested, including the activity of the disease, the type and titre of the ANCA antibodies, the inflammatory status, and the kidneys’ condition.

## 2. Results

### 2.1. AAV Patients Produce Higher Quantities of Cytokines and Chemokines

Among all the assessed biological agents, IL-6 (Figure 1B), IL-8 (Figure 1C), IL-10 (Figure 1D), TNF (Figure 1F), and MIG (Figure 1I) were significantly elevated in AAV patients. At the same time, the level of IL-1β (Figure 1A) was significantly higher in healthy people compared to AAV patients. There was no difference in the serum level of IL-12p70 (Figure 1E), IFN-α (Figure 1G), and IP-10 (Figure 1H).

### 2.2. Haemodialysis Does Not Influence the Level of Cytokines and Chemokines in AAV Patients

Among all the assessed chemicals, there were no differences in the levels when the kidney status was considered. The levels of IL-6 (Figure 2B), IL-8 (Figure 2C), and IL-10 (Figure 2D) were lower in healthy volunteers compared to both CKD and HD patients. The TNF (Figure 2F) level was significantly lower in healthy people than in CKD patients, and the MIG (Figure 2I) concentration was lower in healthy people than in HD patients. On the other hand, IL-1β (Figure 2A) was higher in healthy controls than in CKD patients. Other parameters did not show any significant differences amongst the analysed groups.

### 2.3. Treatment Does Not Affect the Level of Most Chemokines and Cytokines in AAV Patients

Some of the patients recruited for the project were newly diagnosed or being diagnosed for vasculitis; therefore, they did not receive any treatment for their condition. Others, typically those long affected by the disease, received standard cyclophosphamide + glucocorticosteroids (CY + GCS) treatment. The treatment itself did not affect the level of the assessed biologicals, with one exception. The MIG level was significantly higher in non-treated patients than in patients treated with CY + GCS (Figure 3I). The level of TNF was higher in untreated patients compared to healthy people (Figure 3F). IL-6 (Figure 3B), IL-8 (Figure 3C), and IL-10 (Figure 3D) were lower in healthy people than in treated and non-treated patients. On the other hand, the level of IL-1β was higher in healthy people (Figure 3A). There were no differences noted between the assessed groups in the case of IL-12p70 (Figure 3E), IFN-α (Figure 3G), and IP-10 (Figure 3H).

### 2.4. The Activity of the Disease Does Not Affect the Level of Assessed Chemokines and Cytokines

Next, the serum cytokines and chemokines were compared based on the disease activity measured with BVAS. In our analysis, the cut-off was set at 16, where a BVAS < 16 was considered low and a BVAS ≥ 16 was considered high. The differences between low and high BVAS groups are not present, but it is worth remarking that the levels of IL-6 (Figure 4B), IL-8 (Figure 4C), and IL-10 (Figure 4D) were consistently higher in patients, whereas the level of IL-1β (Figure 4A) was lower in patients than in healthy people. In the case of MIG (Figure 4I), the level was higher in patients with a high BVAS than in healthy volunteers, and the levels of IL-12p70 (Figure 4E), TNF (Figure 4F), IFN-α (Figure 4G), and IP-10 (Figure 4H) were not significantly different between the distinguished groups.

### 2.5. The Type of Antibodies Detected in AAV Patients Does Not Affect the Level of the Assessed Chemokines and Cytokines

The patients recruited for our study had either MPO-ANCA or PR3-ANCA antibodies. The type of antibodies did not affect the levels of the studied cytokines and chemokines. Healthy volunteers had higher levels of IL-1β (Figure 5A), but IL-6 (Figure 5B), IL-8 (Figure 5C), and IL-10 (Figure 5D) were lower in this group compared to both patients with MPO-ANCA and PR3-ANCA antibodies. The level of TNF (Figure 5F) was lower in healthy controls than in patients with MPO-ANCA. No significant differences between the studied groups were observed in the case of IL-12p70 (Figure 5E), IFN-α (Figure 5G), IP-10 (Figure 5H), and MIG (Figure 5I).

### 2.6. The Measured Cytokines and Chemokines Correlate with Selected Parameters

The collected data from the AAV patients allowed us to check the possible correlation of the assessed chemokines and cytokines with specific criteria, such as the disease duration at the moment of obtaining blood, BVAS, CRP, eGFR, PR3-ANCA, and MPO-ANCA titres. The levels of IL-6, IL-8, IFN-α, and MIG were positively correlated with the BVAS, and the value of the CRP was positively correlated with IL-6, IL-8, IL-10, IP-10, and MIG. On the other hand, cytokines, such as IL-6, IL-10. and IP-10, and MIG chemokines, negatively correlated with eGFR. Also, worth noting is the positive correlation between PR3-ANCA and the concentration of IFN-α. The *p* and r values for all the analysed parameters are presented in Table 1.

## 3. Discussion

We aimed to assess the level of selected cytokines and chemokines in the serum of AAV patients. The evaluation was performed using flow cytometry. The technique was chosen over others, such as ELISA, because it allows many analytes in the same sample to be measured. That way we eliminate the slight but still existing probability of variations between the collected samples and decreases the amount of material needed to analyse such a wide variety of peptides.

One of analysed cytokines was IL-1β, a pro-inflammatory cytokine, which pre-synthesised pro-IL-1β in both the monocytes/macrophages and neutrophils and, upon the activation of the inflammasome, it becomes activated by the cleavage of caspase-1 and is released from the cell [15]. The fact that it is pre-synthesised allows for quick action when the need arises [16]. Therefore, IL-1β is considered one of the most important cytokines in innate cytokine signalling [17]. The concentration of IL-1β was much higher in our healthy controls than in AAV patients. It may be considered surprising, as long as we only think about IL-1β as a pro-inflammatory cytokine. The cells of AAV patients, who have chronic low-grade inflammation, may be continuously stimulated to release IL-1β, therefore they are not able to accumulate the pro-IL-1β in the cells at high levels. IL-1β was similar in AAV patients with CKD stage 3–4 and HD patients. Also, the treatment, the activity of the disease, or the type of the present antibodies (MPO-ANCA vs. PR3-ANCA) did not influence the serum IL-1β.

After the release of IL-1β, it activates in positive feedback the further production and release of IL-1β, but it also induces the release of IL-6 [17]. IL-6 is considered one of the pro-inflammatory cytokines as well. It is responsible, for e.g., for the induction of CRP production in the liver, which explains why it is positively correlated with the serum CRP level in our AAV patients. It also positively correlates with the activity of the disease measured by the BVAS, which additionally underlines the pro-inflammatory nature of the cytokine itself and the disease [18]. The role of IL-6 in the development of many autoimmune diseases is well-known. There is a known correlation between the level of IL-6 and the severity of such autoimmune diseases as rheumatoid arthritis (RA) [19], systemic lupus erythematosus (SLE) [20], and Crohn’s disease [21]. Although the role of IL-6 in the pathogenesis of large vessel vasculitis is well-defined [22], its possible impact on AAV development and progression is not as well elucidated, but Monach et al. in a relatively new paper indicated that the serum of patients with an active stage of AAV is characterised by a higher concentration of IL-6 than in the serum of patients with an inactive phase [14]. It is suggested that IL-6 can activate neutrophils, which in turn start to express ANCA antigens on their surface. The binding of those receptors results in the release of pre-produced PR3 or MPO, which, after the engulfment by endothelial cells, activate damaging processes, such as cell apoptosis and the production of oxidants [8,10,11]. IL-6 can also cause direct damage to the endothelial cells, and its elevated levels may be linked to a higher risk of relapse and poor prognosis [23], but its level is still typically lower than in the serum of patients with acute ongoing infection, which in turn may be explained as the result of prolonged immune activation and, probably, at least in part, by immunosuppressive treatment [24]. Even though, in the sera of the group of patients treated with CY + GCS, the level of IL-6 was not visibly lower than in the sera of those who were not treated at all, the levels of IL-6 in patients who did not receive any drugs were more diverse, suggesting that the CY + GCS regimen, at least in part, keeps the pro-inflammatory reaction in check, if not inhibiting it completely.

Poor prognosis is linked not only with increased levels of IL-6, but also with elevated IL-8 [23]. This chemokine with evident pro-inflammatory properties is produced as a response to pro-inflammatory stimuli, such as IL-1 or TNF-α [25]. It can be produced by a wide variety of cells, such as renal and endothelial cells, which explains the heavy load of IL-8 in the glomeruli and glomerular capillaries of patients with AAV [26]. IL-8 is also responsible for the attraction of neutrophils [26]. As mentioned above, ANCA-activated neutrophils can produce IL-8 themselves, which explains the relatively poor infiltration of kidneys with activated neutrophils. The intravascular high concentration of IL-8 inhibits neutrophils’ trafficking towards the organ’s tissue [26,27]. Therefore the high level of IL-8 in our AAV patients is not surprising and in agreement with previous data [14,23,28]. The evident positive correlations between disease activity and the CRP level are also logical. The AAV patients with MPO-ANCA antibodies also tend to have higher levels of IL-8 than patients with PR3-ANCA antibodies; the differences are not statistically significant though, which in part may be explained by the relatively small groups of analysed patients.

Another cytokine level, IL-10, tends to be higher in AAV patients than in healthy controls. Similar results were obtained by Wikman et al. [24] and Hoffman et al. [29]. The same as in the case of IL-6, it tends to be even higher in patients with ongoing acute inflammation. In addition to positively correlating with the CRP level, IL-10 negatively correlates with the eGFR level in our patients. It is believed that IL-10, as an anti-inflammatory cytokine, plays a role in counterbalancing chronic autoimmune inflammation in AAV and other inflammatory entities, as it inhibits the Th1-type response and monocytes’ activation [23,30]. Lower levels of IL-10 are seen in the plasma of patients who relapse [23,31]. Some polymorphisms of gene encoding IL-10 were identified in GPA patients as the ones which cause the decreased production of IL-10 and, therefore, increase the risk of relapse [30,31,32]. It was suggested that monitoring patients with low levels of IL-10 might help predict relapse [31].

IL-12p70 is considered the pro-inflammatory cytokine linking innate and adaptive immune systems. Released by macrophages and DCs (dendritic cells), it is responsible for switching Th0 cells towards Th1 cells [33]. Those cells, in turn, start to release IFN-γ as a result. The data regarding the levels of IL-12p70 in ANCA-positive AAV are scarce. In our cohort, the level of this cytokine was very diverse in healthy controls and AAV patients, and none of the analysed parameters had an influence on its level. Other teams reported that, for e.g., the IL-12p70 level is not affected by the presence of MPO-ANCA antibodies [34]. In other autoimmune diseases, an elevated level of IL-12p70 in serum is detected, though, with the example being the generalised myasthenia gravis [35]. The proof that IL-12p70 plays an essential role in selected autoimmune diseases is that anti-IL-12/IL-23p40 biological treatment is used successfully, for e.g., in SLE [36] and Crohn’s disease [37].

TNF is one of the pro-inflammatory cytokines which stimulates the release of IL-1β by neutrophils [38]. What is more, ANCA antibodies can activate TNF-α-primed neutrophils [26]. Those neutrophils degranulate and produce oxidants [8], and other pro-inflammatory cytokines, such as IL-1β mentioned above. Therefore creating a niche promotes vascular damage [38] because both IL-1β and TNF can activate the release of IL-8, which, thanks to its chemokine mode of action, can lead to the infiltration of the vessel wall by other immune cells [26]. That explains the higher TNF level registered in our AAV patients’ serum than in the healthy volunteers’ serum. Moreover, it has been elevated primarily in patients with MPO-ANCA. On the other hand, there are also papers showing that the level of TNF does not differ between AAV patients and healthy people [29]. The role of TNF-α in promoting localised inflammation in the glomeruli of AAV patients was also proven [39]. Then again, anti-TNF therapy used for the treatment of other autoimmune diseases, such as RA or Crohn’s disease, seems to increase the risk of vasculitis development and, apart from a few successful reports, this type of therapy is not generally proposed for this group of patients [40].

Of all the assessed chemokines, only the MIG (CXCL9) levels showed differences in the studied groups. It was higher in AAV patients than in healthy volunteers, especially in those patients before any treatment was employed. MIG is a Th1 cell chemoattractant induced by IFN-γ, released by many immune and non-immune cells, but mainly by monocytes/macrophages. It binds to the CXCR3 receptor expressed preferentially on Th1 cells [41]. Even though there are no known studies assessing the level of MIG in AAV patients’ sera, it is believed that this chemokine might play an important role in the kidneys’ pathology in AAV. It has also been shown that the high number of Tregs expressing CXCR3 can infiltrate the renal tissue, due to the trafficking towards released CXCL9 [42].

Other studied chemokines, IFN-α and IP-10 (CXCL10) did not show any differences in the levels between our studied groups. Even though they were not or were barely detectable in healthy controls, the high diversity in the levels of those studied chemokines in AAV patients did not enable us to find any differences worth noting. The reason behind such results may be the small size of the analysed groups or the lack of actual meaningful differences regarding the level of those chemokines in the pathomechanism of AAV. It is generally believed that IP-10 secretion is linked with Th1 cell activation and over activation. Also, elevated serum IP-10 has been linked with several autoimmune diseases, such as SLE or Graves’ disease [43,44]. IFN-α is considered a Th1 chemokine. It is mainly produced and released by DCs and is considered one of the first in an innate immune response [45]. Unsurprisingly, in some autoimmune diseases, for e.g., the SLE level in the serum is elevated, and anti-IFN-α monoclonal antibody treatment (e.g., sifalimumab) showed promising results [46]. Although we could not prove that IFN-α is higher in AAV patients, we found a correlation with PR3-ANCA levels, which may indicate its role in the development of vasculitis.

### Limitations of the Study

The authors are aware that the performed study has its limitations. The major one being the number of subjects in every analysed group. The fact that the AAV is considered a rare group of autoimmune diseases partially explains this though. Especially difficult was the recruitment of patients without any treatment. Over a 2-year period, we managed to recruit only 11 patients with newly diagnosed AAV in our centre. Anyway, we decided that even though the numbers were not impressive, the results obtained still have merit and are worth presentation.

It should also be mentioned that the concentrations of the assessed chemokines in the sera of the studied groups showed the highest diversity. Even though the concentrations of some of the analysed cytokines were low, in most of the cases they were still detectable, especially in the patients’ sera. In case of chemokines in both AAV patients and healthy volunteers, there were instances of concentrations below the detection thresholds. It is explained, partially at least, by the much higher cut-off values for the studied chemokines than cytokines.

## 4. Materials and Methods

### 4.1. Patients

The blood for cytokines and chemokines assessment was obtained from 20 AAV patients and 7 healthy people. The AAV patients were under the care of professionals from the Nephrology, Transplantology, and Internal Diseases Clinic at the University Clinical Centre, Medical University of Gdańsk, and the Nephrology and Translational Medicine Clinic at the Medical University of Wrocław. Written informed consent was obtained from all enrolled subjects before blood sample collection. The work described here has been carried out following the Declaration of Helsinki. The Local Independent Committee for Ethics in Scientific Research at the Medical University of Gdansk and the Medical University of Wrocław reviewed and approved the experiment protocol and outline (NKBBN/806-254/2021 and pbm199/2016, accordingly).

The full characteristics of the patients and healthy controls are presented in Table 2. The patients enrolled in the study were diagnosed with AAV 25 ± 38 months earlier than when the material was obtained. Overall, 14 of them had chronic kidney disease stages 3–4, and 6 required haemodialysis. A total of 11 patients were newly diagnosed and did not receive any treatment, whereas 9 patients were treated with CY and GCS. Patients’ disease activity was measured using the BVAS (Birmingham Vasculitis Activity Score), where the values below 16 were considered low activity, and the values equal to or more than 16 were considered high activity. A laboratory test included the serum creatinine, estimated glomerular filtration rate (eGFR), C-reactive protein (CRP), PR3-ANCA, and MPO-ANCA.

Healthy volunteers were without any autoimmune, cardiovascular, or neoplastic disease. Every person enrolled in the study provided their written informed consent for obtaining blood and processing.

### 4.2. Methods

#### 4.2.1. Obtaining Material and Storage

A total of 5 mL of venous blood was obtained from each patient, into a tube without any anti-coagulant or preservative. The blood was processed within 30 min of withdrawal. The blood was centrifuged at 4000 rpm for 10 min to separate the serum. Then, the serum was collected in cryogenic tubes and stored at (−80 °C) until usage.

#### 4.2.2. Cytokines and Chemokines Level Assessment

Cytokines were analysed using an inflammatory CBA (cytokine beads array) kit (cat. number 551811, Becton Dickinson, Holdrege, NE, USA). It allows the measurement of 6 different cytokines (IL-12p70, TNF, IL-10, IL-6, IL-1β, and IL-8) in one tube using a flow cytometer. The manufacturer’s protocol was followed. Briefly, 50 µL of sera were added to 50 µL of mixed antibody-coupled beads, each specific for a particular cytokine and conjugated with APC (allophycocyanin) fluorochrome. Then, after 1.5 h incubation at room temperature and protection from light to allow proper binding, the mixture was washed and centrifuged at 1100 rpm for 5 min. After discarding the supernatant, 50 µL of detection reagent was added, incubated for another 1.5 h, and then centrifuged in the same conditions as before. To analyse the specific concentrations of cytokines, the standard curve dilutions for each cytokine were prepared following the same protocol. The cut-off values for the specific cytokines are presented in Appendix A (Table A1).

MIG (CXCL9), IP-10 (CXCL10), and IFN-α were measured using BD CBA Human Flex Sets (catalogue numbers 558286, 558280, 560379, respectively, Becton Dickinson, USA), according to the manufacturer’s protocol. The same as in the case of the cytokines assessment, the concentration of specific chemokines was assessed based on the standard curves. The principles of the technique are the same as in the case of the inflammatory CBA kit. Briefly, 50 µL of serum or standards were mixed with antibody-coupled beads. To allow proper binding, the mixture was incubated for 1 h at room temperature and protected from light. Then, 50 µL of detection reagent was added, followed by 2 h incubation, washing, and centrifugation at 1100 rpm for 5 min. All samples were then analysed using a FACSAria III flow cytometer (Becton Dickinson, USA). The cut-off values for the specific chemokines are presented in Appendix A (Table A1).

#### 4.2.3. Data Analysis

The chemokines and cytokines concentrations were analysed using X = Log(X) transformation and non-linear regression, with least squares regression fitting using GraphPad Prism version 9.5.1. (GraphPad Software, La Jolla, CA, USA).

The Shapiro–Wilk test was used to check whether the continuous variables followed a normal distribution; they did not, therefore, non-parametric tests were used for further analyses. The significance of the differences between the two groups (e.g., HC vs. AAV) was assessed using the Mann–Whitney U test, with *p* < 0.05 considered statistically significant. The difference between the three groups (e.g., HC vs. non-treated patients vs. treated patients) was calculated using the Kruskal–Wallis test, with Dunn’s multiple comparison test. Correlations between certain parameters, such as for e.g., the BVAS, eGFR, and disease duration and cytokines or chemokines levels, were assessed using the nonparametric Spearman’s rank correlation with a two-tailed *p*-value.

The statistical analysis and all the plots were prepared using GraphPad Prism version 9.5.1.

## 5. Conclusions

Due to its rarity, AAV is frequently misdiagnosed as infections, malignancies, depression, or osteoarthritis, especially in older patients. Also, the basic diagnostic scheme relies on clinical features in combination with positive ANCA serology. Importantly, positive ANCA serology can also be found in other conditions with systemic symptoms or asymptomatic patients. Ideally, the diagnosis of AAV is confirmed with a biopsy. We have shown that the serum concentrations of selected cytokines and chemokines in AAV patients are visibly elevated compared to the healthy volunteers. Moreover, some were correlated with disease activity, the CRP level, and eGFR. Most of the studied molecules are implicated in trafficking primed neutrophils towards small vessels, their activation and subsequent degranulation. It seems reasonable to look for links between the levels of these cytokines/chemokines and the clinical symptoms of AAV, which in the future may facilitate the diagnosis and predict the progression of the disease.

## Figures and Tables

**Figure 1 ijms-24-15319-f001:**
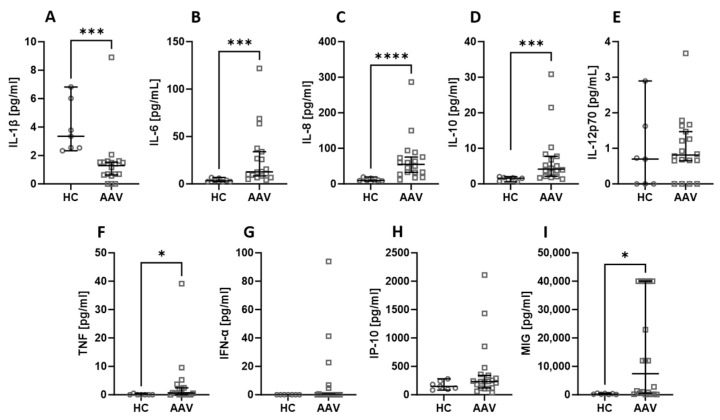
The differences in cytokines and chemokines levels between healthy volunteers (HC) and AAV patients. Median with 95% CI; Mann–Whitney U test; * *p* < 0.05, *** *p* < 0.0005, **** *p* < 0.0001.

**Figure 2 ijms-24-15319-f002:**
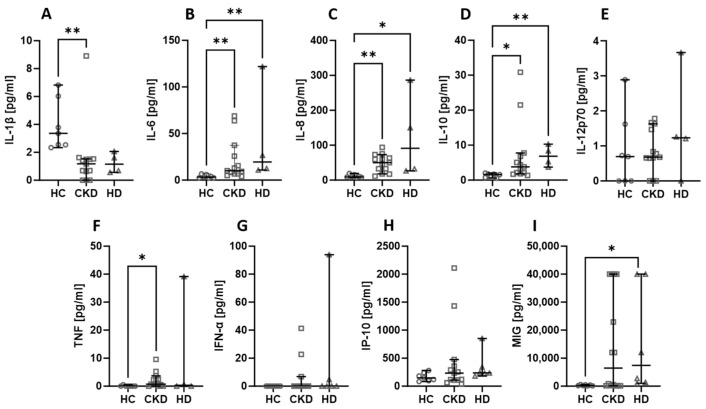
The differences in concentrations of cytokines and chemokines between healthy controls, CKD patients, stages 3–4, and haemodialysis (HD) patients. Median with 95% CI; Kruskal–Wallis test with Dunn’s multiple comparison test; * *p* < 0.05, ** *p* < 0.001.

**Figure 3 ijms-24-15319-f003:**
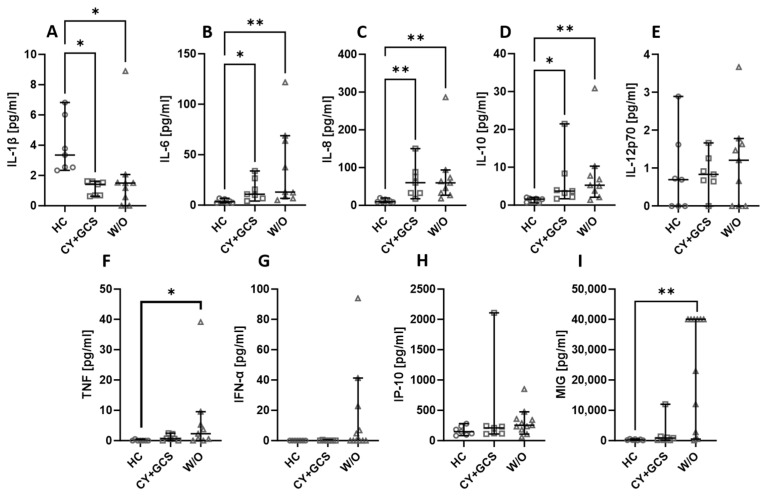
The chemokines and cytokines levels in AAV patients treated with cyclophosphamide and glucocorticosteroids (CY + GCS) and patients without treatment (W/O). Median with 95% CI; Kruskal–Wallis test with Dunn’s multiple comparison test; * *p* < 0.05, ** *p* < 0.001.

**Figure 4 ijms-24-15319-f004:**
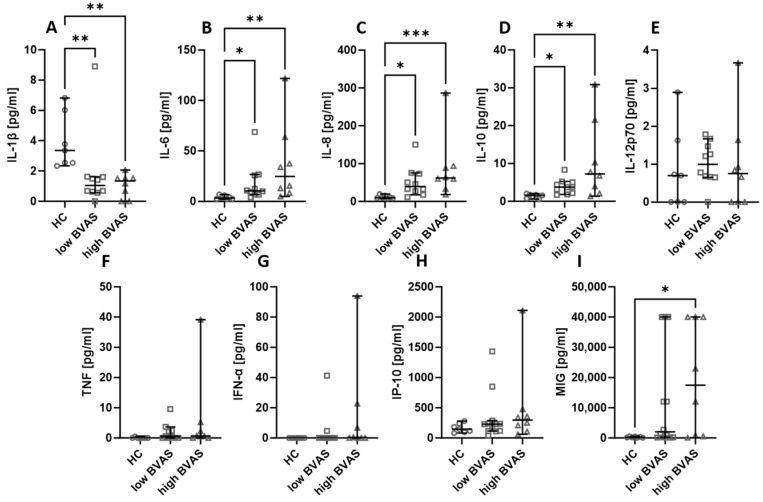
The levels of chemokines and cytokines in the blood of patients with low (<16) and high (≥16) BVAS. Median with 95% CI; Kruskal–Wallis test with Dunn’s multiple comparison test; * *p* < 0.05, ** *p* < 0.001, *** *p* < 0.0005.

**Figure 5 ijms-24-15319-f005:**
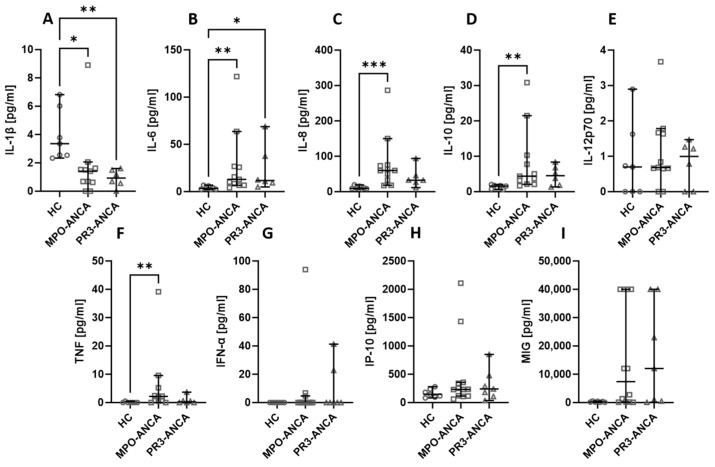
The levels of cytokines and chemokines in MPO-ANCA- and PR3-ANCA-positive patients. Median with 95% CI; Kruskal–Wallis test with Dunn’s multiple comparison test; * *p* < 0.05, ** *p* < 0.001, *** *p* < 0.0005.

**Table 1 ijms-24-15319-t001:** Correlations between the assessed chemokines and cytokines and specific parameters of vasculitis patients.

	Disease Duration	BVAS	CRP	eGFR	PR3-ANCA	MPO-ANCA
IL-1β	*p* = 0.4220	*p* = 0.4907	*p* = 0.7685	*p* = 0.1827	*p* = 0.1443	*p* = 0.2972
	r = (−0.2018)	r = 0.1737	r = 0.07465	r = (−0.3288)	r = 0.4488	r = (−0.5429)
IL-6	*p* = 0.8864	***p* = 0.0451**	***p* = 0.0007**	***p* = 0.0192**	*p* = 0.3160	*p* = 0.9194
	r = (−0.0363)	**r = 0.4775**	**r = 0.7248**	**r = (−0.5455)**	r = 0.3169	r = (−0.0857)
IL-8	*p* = 0.3390	***p* = 0.0170**	***p* = 0.0464**	*p* = 0.0519	*p* = 0.6169	*p* = 0.8028
	r = (−0.2393)	**r = 0.5541**	**r = 0.4750**	r = (−0.4649)	r = 0.1620	r = 0.1429
IL-10	*p* = 0.8703	*p* = 0.1154	***p* = 0.0492**	***p* = 0.0037**	*p* = 0.9380	*p* = 0.5717
	r = (−0.0414)	r = 0.3843	**r = 0.4698**	**r = (−0.6467)**	r = (−0.02817)	r = (−0.0092)
IL-12p70	*p* = 0.5513	*p* = 0.8568	*p* = 0.3588	*p* = 0.0704	*p* = 0.7091	*p* = 0.0778
	r = (−0.1504)	r = 0.04581	r = 0.2299	r = (−0.4361)	r = 0.1199	r = (−0.7827)
TNF	*p* = 0.3170	*p* = 0.4504	*p* = 0.4455	*p* = 0.7430	*p* = 0.2291	*p* = 0.667
	r = (−0.2500)	r = 0.1899	r = 0.1919	r = (−0.0831)	r = 0.3759	r = (−0.8197)
IFN-α	*p* = 0.2749	***p* = 0.0242**	*p* = 0.1032	*p* = 0.4936	***p* = 0.0242**	*p* = 0.5717
	r = (−0.2565)	**r = 0.5017**	r = 0.3751	r = (−0.1625)	**r = 0.5017**	r = (−0.2673)
IP-10	*p* = 0.6035	*p* = 0.1780	***p* = 0.0019**	***p* = 0.0202**	*p* = 0.3807	*p* = 0.3536
	r = 0.1236	r = 0.3137	**r = 0.6506**	**r = (−0.5149)**	r = (−0.2652)	r = 0.4286
MIG	*p* = 0.3635	***p* = 0.0381**	***p* = 0.0002**	***p* = 0.0039**	*p* = 0.8710	*p* = 0.8873
	r = (−0.2146)	**r = 0.4665**	**r = 0.7340**	**r = (−0.6146)**	r = (−0.05049)	r = 0.07207

**Table 2 ijms-24-15319-t002:** Characteristics of AAV patients and healthy volunteers. The data are presented as the median (interquartile range, IQR).

	Vasculitis Patients (AAV)			Healthy Control (HC)
	All	Treated	w/o Treatment	
Group size	20	9	11	7
Gender (F/M)	7/13	4/5	3/8	5/2
Age	61 (13.25)	60 (11)	61 (13)	71 (4.5)
Disease duration (months)	15 (19.5)	16 (23)	8 (17)	---
eGFR (mL/min/1.73 m^2^)	22 (33.5)	25 (23)	15 (26)	---
CKD */HD	14/6	7/2	7/4	---
Treatment regimen (w/o treatment/CY + GCS)	11/9	0/9	11/0	---
BVAS	15 (9.25)	6 (8)	17 (5)	---
Low BVAS/high BVAS **	12/8	7/2	5/6	---
CRP (mg/L)	18 (39.5)	8 (18)	36 (37)	---
C-ANCA positive patients (F/M)	13 [5/8]	7 [3/4]	6 [2/4]	---
C-ANCA titre	123 (132.12)	68 (104)	198 (63)	---
P-ANCA positive patients (F/M)	7 [2/5]	2 [1/1]	5 [1/4]	---
P-ANCA titre	124 (132)	40 (19)	164 (55)	---

* Stage 3 or 4, ** low BVAS < 16, high BVAS ≥ 16. CKD—chronic kidney disease, HD—haemodialysis, CY—cyclophosphamide, GCS—glucocorticosteroids, BVAS—Birmingham Vasculitis Activity Score, CRP—C-reactive protein, eGFR—estimated glomerular filtration rate, C-ANCA—antineutrophil cytoplasmic antibodies (~anti-proteinase-3 antibody, PR3-ANCA), P-ANCA—antineutrophil perinuclear antibodies (~anti-myeloperoxidase antibody, MPO-ANCA); data are presented as the median (interquartile range, IQR).

## Data Availability

The data presented in this study are available upon request from the corresponding author.

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
