# Peer review of "Chemokines and Cytokines Profiles in Patients with Antineutrophil Cytoplasmic Antibodies-Associated Vasculitis: A Preliminary Study"

_ijms, 2023, doi:10.3390/ijms242015319_

Round 1

Reviewer 1 Report

In this study, pro-inflammatory levels of cytokines/chemokines were assessed in 20 ANCA vasculitis patients (and 7 controls).

The major limitations of the study is the very low number of patients included, precluding any high powered analyses.

Moreover, only 11 patients were treatment naïve (and 9 patients were still on a unclear treatment regimen).

Another major limitation of the study is the results presentation, raw data must be displayed when reporting such low number of patients.

Is this state, i do not recommend publishing such data.

However, because the scientific question is of interest for the general audience, I would consider reviewing the paper again 

- if the number of subject is significantly improved.

- and if data presentation (especially clinical characteristic at serum analysis) is clearly improved.

My remarks are detailed bellow (more may come in a second reviewing of the paper):

In the abstract, is missing :

- number of subject included

- timing of serum evaluation (at diagnosis or after)

- technique of assessment

Introduction:

- "biological treatment is a relatively new approach" cannot be said (MAINRITSAN1 = 2014 ; RITUXVAS/RAVE = 2010)

- rituximab is now the SoC for maintenance (and nearly the prefered approches for induction): no patients seem to have been treated with RTX in this study?

- the section is a bit too long and should be reduced to focus on the study main interest: potential role of cytokines in AAV pathogenesis, potential impact on clinical practice to evaluate those levels

Methods :

- major limitation of the study: very small number of patients (too small to make any conclusion)

- considering the total number of patients, medians [IQR] should be used to present data (table 2) and not mean +/- SD

- not clear if ANOVA or kruskall wallis test was used for 3 groups comparison

- not clear why mean disease duration is 25 months whereas 11 patients where newly diagnosed (9 patients still receiving CY + GCS after more than 25 months?)

- a more precise description of patients if needed, especially considering organ involvement and therapeutic management (cyclophosphamide only? no rituximab?)

- suggest describing "all patients" "no treatment patients" "undertreatment patients" in table 2

- was all patients analysed on the same day on the cytometer of on different days? if on different days, how data was compared? (positive control?)

- the use of BVAS cutoff at 16 should be justified

- why were cytokines log transformed? it does not appear to be log transformed on the axis? is it because of the method (flow cytometry?)

- catalog number of kit used for cytokine quantification should be provided

- at which timepoint BVAS was evaluated? at diagnosis or at serum assessment?

Results :

- all figures: suggest using dotplot and show raw data (rather than box plot alone)

- how many patients had levels below detection limits for each cytokine?

- sensibility treshold for each cytokines should be provided (as a supplementary data)

- Table 1 : a correlation plot with raw data should be provided instead (and also correlation between cytokine could be of interest)

- all figures: suggest using colors, to enhance readability

Discussion:

- a limitation section is lacking

- the justification of flow cytometry (rather than ELISA or multiplex) for cytokine assessment is lacking

- number of patients with below treshold data should be discussed

Author Response

Thank You for the extensive reviewing of the paper. Your suggestions were valuable and the implementation of them into the paper has increased its' value. Please find below our point-by-point response.

“In this study, pro-inflammatory levels of cytokines/chemokines were assessed in 20 ANCA vasculitis patients (and 7 controls).

The major limitations of the study is the very low number of patients included, precluding any high powered analyses.

Moreover, only 11 patients were treatment naïve (and 9 patients were still on a unclear treatment regimen).

Another major limitation of the study is the results presentation, raw data must be displayed when reporting such low number of patients. Is this state, i do not recommend publishing such data.

However, because the scientific question is of interest for the general audience, I would consider reviewing the paper again 

- if the number of subject is significantly improved.

- and if data presentation (especially clinical characteristic at serum analysis) is clearly improved.”

Thank you for the comment. We realize that the patient number is relatively low. However, ANCA-associated vasculitis is classified as an orphan disease under the number ORPHA:156152 on Orphanet (https://www.orpha.net/consor/cgi-bin/OC_Exp.php?lng=EN&Expert=156152). Thanks to the POLVAS registry, it was possible to detect only 625 patients with confirmed AAV in Poland (data from 2018, https://link.springer.com/article/10.1007/s10067-019-04538-w). Especially difficult is the recruitment of patients without any treatment. AAV is frequently misdiagnosed as infections, malignancies, depression, or osteoarthritis, especially in older patients. Due to the variety of affected organs and heterogeneous symptoms, overlapping other diseases, AAV can remain undiagnosed for months or years until ANCA testing is performed. As initial clinical presentations are diverse and often non-specific, AAV is an infrequent but important differential diagnosis for many conditions across many medical disciplines. Given the rarity of AAV and the existence of diseases that mimic vasculitis, the diagnosis should be reviewed periodically, especially in cases of inadequate response to treatment or inconsistency of some of the disease manifestations with AAV. Over a 2-year period, we managed to recruit only 11 patients with newly diagnosed AAV in our center. We added this information in the section “Limitations of the study”. We also propose changing the title of the manuscript into “Chemokines and cytokines profiles in patients with ANCA-associated vasculitis - a preliminary study.”

The information about the treatment of the rest patients was given in the Result section (“2.3. Treatment does not affect the level of most chemokines and cytokines in AAV patients”; “. Others, typically long-affected by the disease, received standard cyclophosphamide + glucocorticosteroids (CY+GCS) treatment.”, page 4, lines 136-138) and Method section as well (“14 of them had chronic kidney disease stages 3-4, and 6 required haemodialyses. 11 patients were newly diagnosed and did not get any treatment, whereas 9 patients were treated with CY and GCS.”, page 9, lines 341-343).

“In the abstract, is missing :

- number of subject included”

The number of patients and healthy controls were included in the abstract (line 16).

“- timing of serum evaluation (at diagnosis or after)”

The timing of the cytokines and chemokines assessment was included in lines 16-17.

“- technique of assessment”

The technique used to assess the concentration of the analysed chemicals is now mentioned in line 14.

The new version of the abstract is as follows (page 1 of the manuscript, lines 13-26):

“The damage of small vessels in AAV and inflammatory reactions are accompanied by the release of various chemokines and cytokines. Using flow cytometry technique, we assessed the levels of specific cytokines: IL-1β IL-6, IL-8, IL-10, IL12p70 and TNF, and chemokines: IFN-α, IP-10, MIG in the serum of 9 healthy volunteers and 20 AAV patients where 11 of patients were not treated and evaluated at the time of diagnosis and 9 were already diagnosed and taking CY+GCS. The obtained results were then compared considering the activity of the disease, the ANCA antibodies type and titre, inflammatory status and the kidneys’ condition. Amongst others, IL-6, IL-8, IL-10, TNF, and MIG levels were much higher in the serum of AAV patients than in healthy controls, whereas the level of IL-1β was higher in healthy volunteers. Additionally, the levels of IL-6, IL-10, IP-10 and MIG negatively correlated with eGFR level, while the level of IFN-α positively correlated with the titre of PR3-ANCA. As most of those molecules are implicated in trafficking primed neutrophils towards small vessels, looking for links between the levels of these cytokines/chemokines and clinical symptoms of AAV may facilitate the diagnosis and predict the progression of the disease.”

“Introduction:

- "biological treatment is a relatively new approach" cannot be said (MAINRITSAN1 = 2014 ; RITUXVAS/RAVE = 2010)”

Thank you for the comment. We removed this sentence from the Introduction.

“- rituximab is now the SoC for maintenance (and nearly the prefered approches for induction): no patients seem to have been treated with RTX in this study?”

Amongst the patients enrolled in the study, no one was treated with RTX at the time. In a previous year, we started a group of patients on RTX but their sera were not analysed for the concentrations of cytokines and chemokines yet.

“- the section is a bit too long and should be reduced to focus on the study main interest: potential role of cytokines in AAV pathogenesis, potential impact on clinical practice to evaluate those levels”

To increase clarity and emphasize the main topic of the article, a significant part of the introduction has been redacted. We removed lines 53-69 (page 2) and the paragraph starting in line 53 was merged with the paragraph starting in line 70.

“Methods :

- major limitation of the study: very small number of patients (too small to make any conclusion)

- considering the total number of patients, medians [IQR] should be used to present data (table 2) and not mean +/- SD”

The data were changed into median (IQR).

“- not clear if ANOVA or kruskall wallis test was used for 3 groups comparison”

The three groups were compared using the Kruskal-Wallis test, which is also named one-way ANOVA on ranks. In order not to confuse the concepts for readers, we have corrected the name of the test (page 11, line 399).

“- not clear why mean disease duration is 25 months whereas 11 patients where newly diagnosed (9 patients still receiving CY + GCS after more than 25 months?)”

The disease duration was described as the moment of first symptoms. In the case of non-treated patients, it took up to 24 months (in two cases) to confirm a diagnosis and start the treatment (and at the same time to obtain the blood sample). That is why the mean disease duration is so long. When it comes to the treated patients, the disease duration does not mean they were treated with CY + GCS for so long. The treatment they got is the treatment at the time of obtaining the blood sample.

“- a more precise description of patients if needed, especially considering organ involvement and therapeutic management (cyclophosphamide only? no rituximab?)”

As stated in the paper, patients enrolled in the study were either not treated (directly before starting the treatment) or on CY + GCS. No patients enrolled in the study were treated with RTX.

“- suggest describing "all patients" "no treatment patients" "undertreatment patients" in table 2”

The table was redesigned. It looks like the one below right now.

Vasculitis patients (AAV)

Healthy control (HC)

All

treated

w/o treatment

Group size

20

9

11

7

Gender [F/M]

7/13

4/5

3/8

5/2

Age

61 (13.25)

60 (11)

61 (13)

71 (4.5)

Disease duration [months]

15 (19.5)

16 (23)

8 (17)

---

eGFR [mL/min/1.73m2]

22 (33.5)

25 (23)

15 (26)

---

CKD*/HD

14/6

7/2

7/4

---

Treatment regimen

[w/o treatment/CY+GCS]

11/9

0/9

11/0

---

BVAS

15 (9.25)

6 (8)

17 (5)

---

low BVAS / high BVAS**

12/8

7/2

5/6

---

CRP [mg/L]

18 (39.5)

8 (18)

36 (37)

---

C-ANCA positive patients [F/M]

13 [5/8]

7 [3/4]

6 [2/4]

---

C-ANCA titre

123 (132.12)

68 (104)

198 (63)

---

P-ANCA positive patients [F/M]

7 [2/5]

2 [1/1]

5 [1/4]

---

P-ANCA titre

124 (132)

40 (19)

164 (55)

---

“- was all patients analysed on the same day on the cytometer of on different days? if on different days, how data was compared? (positive control?)”

All flow cytometric data were acquired on the same day.

“- the use of BVAS cutoff at 16 should be justified”

The cut-off value for high BVAS was chosen based on the literature. The value ≥16 is generally considered an active AAV. Here are some examples of the papers:

https://link.springer.com/article/10.1007/s10067-011-1838-7

https://academic.oup.com/rheumatology/article/61/12/4603/6549520

“- why were cytokines log transformed? it does not appear to be log transformed on the axis? is it because of the method (flow cytometry?)”

The way of analysis was described in paragraph “4.2.3 Data analysis” (page 11, lines 391-393).

“- catalog number of kit used for cytokine quantification should be provided”

The catalogue numbers have been added (page 10, line 368 for CBA, and page 11, line 380 for FlexSets).

“- at which timepoint BVAS was evaluated? at diagnosis or at serum assessment?”

BVAS was calculated during the visit when the blood was obtained for the tests.

“Results :

- all figures: suggest using dotplot and show raw data (rather than box plot alone)”

The figures were changed according to the suggestion.

“- how many patients had levels below detection limits for each cytokine?”

It all depends on the type of cytokine. As the figures were changed to present the raw data, it is more visible right now.

“- sensibility treshold for each cytokines should be provided (as a supplementary data).”

The cut-off values for each cytokine and chemokine have been added as Table A1 (page 12), and right now it looks like the one below.

[pg/ml]

IL-1β

2.76

IL-6

2.86

IL-8

6.47

IL-10

3.03

IL-12p70

2.81

TNF

2.91

IFN-α

91.20

IP-10

102

MIG

102

“- Table 1: a correlation plot with raw data should be provided instead (and also correlation between cytokine could be of interest)”

We decided to present the correlations in the form of a table, because of the amount of data (figures) it would require to implement into the paper. Even if we choose to present only those parameters which correlate with the highest number of analytes (eGFR, CRP and BVAS) it means an additional 3, quite big figures. As an example please find below the figure presenting the correlations between the concentrations of specific cytokines and chemokines and BVAS.

“- all figures: suggest using colors, to enhance readability”

The figures from 1 to 5, presenting raw data are right now in a grey-black colour palette.

“Discussion:

- a limitation section is lacking”

The limitations of the study were added as a separate paragraph,  directly after the Discussion (page 9, lines 312 - 326).

“- the justification of flow cytometry (rather than ELISA or multiplex) for cytokine assessment is lacking”

The justification for the use of flow cytometry is added to the Discussion part (page 7, lines 191-195)

“- number of patients with below treshold data should be discussed”

The comment about the results below the detection thresholds was added (paragraph “Limitations of the study”, page 9, lines 312-326).

Reviewer 2 Report

Literature search is insufficient.

Line 97 to 99 ‘However, the cytokine profile in AAV is not very well described.’

I don’t agree with the idea. There have been many studies investigating related cytokines cytokines/chemokines in AAV.

ï‚ž           Monach PA, et al. Serum Biomarkers of Disease Activity in Longitudinal Assessment of Patients with ANCA-Associated Vasculitis. ACR Open Rheumatol. 2022;4(2):168-176.

ï‚ž           Monach PA, et al. Serum proteins reflecting inflammation, injury and repair as biomarkers of disease activity in ANCA-associated vasculitis. Ann Rheum Dis. 2013;72(8):1342-1350.

ï‚ž           Tomizawa K, Nagao T, Kusunoki R, et al. Reduction of MPO-ANCA epitopes in SCG/Kj mice by 15-deoxyspergualin treatment restricted by IgG2b associated with crescentic glomerulonephritis. Rheumatology (Oxford). 2010;49(7):1245-1256.

ï‚ž           Kyurkchiev D, Yoneva T, Yordanova A, et al. Alterations of serum levels of plasminogen, TNF-α, and IDO in granulomatosis with polyangiitis patients. Vascular. 2021;29(6):874-882.

ï‚ž           Uno K, Muso E, Ito-Ihara T, et al. Impaired HVJ-stimulated Interferon producing capacity in MPO-ANCA-associated vasculitis with rapidly progressive glomerulonephritis lead to susceptibility to infection. Cytokine. 2020;136:155221.

Line 265: The reference 40 (Hoffmann et al) described not only TNF, but many other cytokines. Please cite correctly with respect to the precursors’ studies.

The review, Int. J. Mol. Sci. 2020, 21, 7319; doi:10.3390/ijms21197319 (Reference 4) has already comprehensively summarized cytokines in AAV.

The cytokines/chemokines investigated in the submitted manuscript have been covered by the above preceding studies.

The topics have already been well-studied and the patient number in the submitted manuscript is smaller than preceding studies. Therefore, the significance as publishing this as a new paper is not clear.

Author Response

Thank You for reviewing our paper. Please find below the response to Your suggestions.

“Literature search is insufficient.

Line 97 to 99 ‘However, the cytokine profile in AAV is not very well described.” I don’t agree with the idea. There have been many studies investigating related cytokines/chemokines in AAV. 

  • Monach PA, et al. Serum Biomarkers of Disease Activity in Longitudinal Assessment of Patients with ANCA-Associated Vasculitis. ACR Open Rheumatol. 2022;4(2):168-176.
  • Monach PA, et al. Serum proteins reflecting inflammation, injury and repair as biomarkers of disease activity in ANCA-associated vasculitis. Ann Rheum Dis. 2013;72(8):1342-1350.
  • Tomizawa K, Nagao T, Kusunoki R, et al. Reduction of MPO-ANCA epitopes in SCG/Kj mice by 15-deoxyspergualin treatment restricted by IgG2b associated with crescentic glomerulonephritis. Rheumatology (Oxford). 2010;49(7):1245-1256.
  • Kyurkchiev D, Yoneva T, Yordanova A, et al. Alterations of serum levels of plasminogen, TNF-α, and IDO in granulomatosis with polyangiitis patients. Vascular. 2021;29(6):874-882.
  • Uno K, Muso E, Ito-Ihara T, et al. Impaired HVJ-stimulated Interferon producing capacity in MPO-ANCA-associated vasculitis with rapidly progressive glomerulonephritis lead to susceptibility to infection. Cytokine. 2020;136:155221.

We agree with the Reviewer that there are different articles describing changes in cytokines/chemokines in AAV patients. The problem is that it is difficult to compare the results due to using different methods to establish cytokine levels, different materials used in the studies (serum vs. plasma), different study models (time of observation), and different study groups (only treated patients, only patients without treatment etc.). Therefore, we aimed to assess the levels of selected cytokines and chemokines taking into account various factors that could influence the levels of the cytokines tested, including the activity of the disease, the ANCA antibodies type and titre, inflammatory status and the kidneys’ condition.

We corrected our statement in the Introductions and cited some articles suggested by the Reviewer (page 3, lines 100-110). We still believe that it is worth publishing the newest articles trying to compare different cytokines depending on AAV patients' status and discuss the results with other authors.

The proposed article by Monach et al. from 2022 was added to the discussion of the obtained result (as the reference no 14, and other positions were changed accordingly. Because the other paper by Monach (from 2013 does not really discuss the analytes we assessed, we decided not to implement it into the paper, even though the paper itself is interesting.

Actually, in the article by Tomizawa et al., the authors use the animal model, which cannot be applied to human studies due to some differences in the immunology between our species. We tried to avoid citing studies using cell lines or animal studies. The article by Uno et al. uses a different study model, that is HVJ-stimulation of WBC, which does not reflect the serum status of this cytokine.

“Line 265: The reference 40 (Hoffmann et al) described not only TNF, but many other cytokines. Please cite correctly with respect to the precursors’ studies.”

 We agree with this opinion. Therefore, we cited this article when describing other cytokines that were shown by these authors. The authors stated that “The following pro-inflammatory cytokines were quantified: IL-1β, IL-6, IL-17 A, IL-17 F, IL-21, IL-22, IL-23, TNF-α, and sCD40L. Anti-inflammatory cytokines included in the analyses were IL-4, IL-10, IL-25, IL-31, IL-33, and INF-γ. However, numerous cytokines could not finally be incorporated in the study since at least 50% of all quantified serum concentrations fell below the detection limit of the assay. Nevertheless, specific cytokines which were detected in such low levels in AAV were significantly elevated in systemic sclerosis: IL-4, IL-6, IL-17A, and IL-22. Thus, the following mediators will not be discussed in AAV: IL-1β, IL-4, IL-6, IL-17A, IL-17F, IL-21, IL-22, IL-23, IL-25, IL-31, INF-γ. Regarding AAV we will mainly address IL-10 (62% of all measured values within range), IL-33 (79% of all measured values within range), sCD40L…” They received similar results concerning IL-10, which we added in the Discussion (page 8, line 250-251) . The authors also studied IL-33 and sCD40L, which we did not examine, so we cannot discuss these results.

“The review, Int. J. Mol. Sci. 2020, 21, 7319; doi:10.3390/ijms21197319 (Reference 4) has already comprehensively summarized cytokines in AAV. The cytokines/chemokines investigated in the submitted manuscript have been covered by the above preceding studies.”

 As written above, we agree with the Reviewer that there are different articles describing changes in cytokines/chemokines in AAV patients and explained that it is worth publishing the newest articles trying to compare different cytokines depending on AAV patients' status (including the activity of the disease, the ANCA antibodies type and titre, inflammatory status and the kidneys’ condition).

The topics have already been well-studied and the patient number in the submitted manuscript is smaller than preceding studies. Therefore, the significance as publishing this as a new paper is not clear.”

Thank you for the comment. We realize that the patient number is relatively low. However, ANCA-associated vasculitis is classified as an orphan disease under the number ORPHA:156152 on Orphanet (https://www.orpha.net/consor/cgi-bin/OC_Exp.php?lng=EN&Expert=156152). Thanks to the POLVAS registry, it was possible to detect only 625 patients with confirmed AAV in Poland (data from 2018, https://link.springer.com/article/10.1007/s10067-019-04538-w). Especially difficult is the recruitment of patients without any treatment, even though we recruited them in two large Polish Medical Centres – in GdaÅ„sk and WrocÅ‚aw. AAV is frequently misdiagnosed as infections, malignancies, depression, or osteoarthritis, especially in older patients. Due to the variety of affected organs and heterogeneous symptoms, overlapping other diseases, AAV can remain undiagnosed for months or years until ANCA testing is performed. As initial clinical presentations are diverse and often non-specific, AAV is an infrequent but important differential diagnosis for many conditions across many medical disciplines. Given the rarity of AAV and the existence of diseases that mimic vasculitis, the diagnosis should be reviewed periodically, especially in cases of inadequate response to treatment or inconsistency of some of the disease manifestations with AAV. Over a 2-year period, we managed to recruit only 11 patients with newly diagnosed AAV in our center. We added this information in the “Limitations of the study” paragraph (page 9, lines 313 – 326). We also propose changing the title of the manuscript to “Chemokines and cytokines profiles in patients with ANCA-associated vasculitis - a preliminary study.” AAV is a rare disease with complicated immunopathology. We believe that our results can be of interest to the scientists interested in this topic.

Round 2

Reviewer 1 Report

While the overall quality of the manuscript has been improved, the number of subjects in the study has not been increased, especially regarding treatment naive patients.

In this form, the paper does not add much to the litterature.

Thus I suggest to not publish this paper in its current form.

Reviewer 2 Report

(There are no comments)